# Correlation analysis of serum TLR4 protein levels and *TLR4 gene* polymorphisms in gouty arthritis patients

**Lu Liu**[1,2☯], **Shuang He**[3☯], **Lin Jia**[1], **Hua Yao**[1], **Dan Zhou**[1], **Xiaobin Guo**[1]*, **Lei Miao**[2]*

**1** The First Affiliated Hospital of Xinjiang Medical University, Xinjiang, China, **2** School of Public Health, Xinjiang Medical University, Xinjiang, China, **3** Gansu Provincial Center for Disease Control and Prevention, Lanzhou, China

☯ These authors contributed equally to this work.
* 506322435@qq.com (XG); 643444508@qq.com (LM)

## Abstract

### Objective

The Toll-like receptor (TLR) 4-mediated nuclear factor kappa B (NF-κB) signaling pathway regulates the production of inflammatory factors and plays a key role in the pathogenesis of gouty arthritis. The aim of the present study was to investigate the link among *TLR4 gene* polymorphisms at various loci, protein expression, and gouty arthritis susceptibility.

### Methods

Between 2016 and 2021, a case-control study was used to collect a total of 1207 study subjects, including 317 male patients with gouty arthritis (gout group) and 890 healthy males (control group). The association between gout susceptibility and different genetic models was analyzed by typing three loci of the *TLR4 gene* (rs2149356, rs2737191, and rs10759932) using a multiplex point mutation rapid assay, and the association between protein expression and gout was confirmed by measuring TLR4 protein concentrations using enzyme-linked immunosorbent assays (ELISAs).

### Results

In a codominant models AA and AG, the rs2737191 polymorphism in the gout group increased the risk of gout compared to the AA genotype (*OR = 2.249, 95%CI 1.010~5.008*), and the risk of gout was higher for those carrying the G allele compared to the A allele (*OR = 2.227, 95%CI 1.006~4.932*). TLR4 protein expression was different between the two groups with different locus genotypes. The differences in TLR4 protein expression between the gout group and control group were statistically significant between the following genotypes: the GG and GT genotypes of the rs2149356 polymorphism; the AA and AG genotypes of the rs2737191 polymorphism; and the TT and TC genotypes of the rs10759932 polymorphism(*P<0.05*). The *TLR4 protein* level in the gout group (19.19±3.09 ng/ml) was significantly higher than that in the control group (15.85±4.75 ng/ml).

**Funding:** Natural Science Foundation of Xinjiang (2022D01C756,2020D01C263), Special fund project of youth scientific Research sailing from The First Affiliated Hospital of Xinjiang Medical University (2022-74), Natural Science Foundation of China (81460153, 82002276). Lu Liu and Xiaobin Guo performed the data analysis and statistical analysis, Xiaobin Guo and Lei Miao conceived the study and projected administration. In addition, Lu Liu participated in the drafting of the manuscript, conducted the epidemiological survey and collected data from molecular biology experiments.

**Competing interests:** The authors have declared that no competing interests exist.

## Conclusion

The AG genotype of the *TLR4 gene* rs2737191 polymorphism may be correlated with the development of gouty arthritis. The level of *TLR4 protein* expression is significantly higher in patients with gouty arthritis than in controls, and there is a correlation between high TLR4 protein expression and the development of gouty arthritis.

## 1. Introduction

When the body consumes foods containing purines under normal conditions, the purines are metabolized in the body to form uric acid (UA), which is present in the plasma in the form of urates. The main site of UA metabolism is in the kidneys. Humans cannot break down small molecules that are not absorbed and digested by the body, such as $NH_3$, and they cannot further degrade UA into soluble allantoin. Therefore, consuming too many purine-rich foods or having a purine excretion issue may lead to the body acquiring a high UA condition. The joint cavities, cartilage, and even kidneys of the body accumulate monosodium urate crystals due to the increased UA content, which results in inflammatory illness with tissue damage and gout attacks. The clinical signs of gout include acute or chronic gouty arthritis, joint deformities, gout stones, and even kidney buildup. Gout symptoms include not only acute pain, chronic pain, functional disability, and irreversible joint damage but also the possibility of major long-term diseases that can create a significant financial burden on families [1]. With the improvement of living standards and changes in diets, epidemiological surveys have shown that the prevalence of gout in the US population has increased from 2.64% in the 1980s [2] to 3.76% in 2010 [3] and 3.90% in 2016 [4] with reported prevalences of approximately 2.49% in the UK in 2015 [5], 5.2% in Australia [5], 1.4% in Germany [6], and 4.75% in Greece [7]. The total prevalence of gout in China's coastal areas and Shenzhen, China was 0.36% and 2.80% in 2002, respectively [8], while the crude prevalence of gout among Jilin University students in China was 13.18% in 2016 [9]. According to a survey conducted in 27 provinces countrywide in 2016, the ratio of men to women with gout in China was approximately 15:1 [10]. Gout prevalence is approximately 0.10–10% worldwide with variances between areas and ethnic groups [11], indicating that gout prevalence is increasing year by year, and gout is affecting younger individuals worldwide [12]. Although the hyperuricemic state is the biochemical basis of gout attacks and the risk of gout increases with increasing serum uric acid (SUA) levels in the body [13], D Dalbeth N et al. [12], also discovered that approximately 90% of patients with hyperuricemia do not progress to gout but instead remain in the hyperuricemic state. These findings indicate that hyperuricemia is only one of the underlying factors in the development of gout and that genetic susceptibility may also play a role. In addition to investigating hyperuricemia-related studies, the etiology and variables impacting the illness should be investigated from the perspective of multiple components operating together in the disease as well as the involvement of immunological inflammatory pathways in gout.

In terms of genetics, there are many pathways involved in regulating or influencing the development of gout, such as the Cytoplasmic nucleotide-binding oligomerization domain-like receptor protein 3 (NLRP3) inflammatory route, Toll-like receptor 4 (TLR4)/MyD88 signaling system, and TLR4-NF-κB signaling pathway. TLR is the upstream signal of NF-κB in the TLR inflammatory pathway and is a receptor for the lipid-based pathogen-associated molecular pattern (PAMP) [14], which is an important pattern recognition receptor in the immune system, and it is present in various immune cells (T cells, B cells, and natural killer

cells) as well as in monocytes, macrophages, neutrophils, and epithelial cells. TLRs are instrumental in the control of adaptive immunity impacting both T and B cell responses, TLR activation provides information about the type of invading pathogen being recognized and instructs lymphocytes to initiate the most appropriate effector response to clear the infection [15]. In recent years, it has been demonstrated that the TLR2/TLR4-NF-κB signaling pathway can be exploited as a target to reduce TLR2/TLR4 protein expression levels to cure gouty arthritis symptoms [16]. TLR activation by exogenous or endogenous ligands has also been shown to cause a signal transduction cascade that secretes pro-inflammatory factors, resulting in an immune inflammatory response [17]. For example, in patients with gouty arthritis, the release of urate crystals from damaged cells is an endogenous danger signal, thus indicating a link between gout pathogenesis and TLRs, this endogenous TLR4 ligand, released by activated macrophages, perpetuates inflammatory response [18,19].

TLR4 is one of the most important family members of the TLR inflammatory pathway and is widely expressed on the surface of various lymphocytes, and activated TLR4 induces the production of a range of inflammatory mediators and regulates cellular expression of pro-inflammatory factors and tissue inflammatory responses to injury [20]. TLR-related genes have been linked to immunological illnesses, but little is known about their etiology and the factors that influence gout [14,21,22]. Our research team has investigated gouty arthritis in terms of gene polymorphisms, and we have identified the following important polymorphisms related to gouty arthritis: rs2231142 and rs72552713 polymorphisms in the ATP-binding cassette transporter subfamily G member 2 gene (*ABCG2 gene)*; rs7349721 polymorphism in the solute carrier family 2 member 9 gene (*SLC2A9 gene)*; rs3806268 and rs10754558 polymorphisms in the *NLRP3 gene*; and rs11935252, rs369028698, and rs3804100 polymorphisms in the *TLR2 gene*. Based on the previous research on gene polymorphisms, in the present study, we evaluated the expression of plasma TLR4 protein in patients with gouty arthritis by enzyme-linked immunosorbent assay (ELISA) and analyzed the correlation between *TLR4 gene* polymorphisms and protein expression differences and gouty arthritis.

## 2. Methods

### 2.1 Patients

From 2016 to 2021, 317 patients with gouty arthritis attending the First Affiliated Hospital of Xinjiang Medical University, the Fourth Affiliated Hospital, or the People's Hospital of Xinjiang Uygur Autonomous Region as well as 890 patients who underwent health examinations in the same period in the above three hospitals were enrolled in the present study. The 317 cases of gouty arthritis met the classification criteria proposed by the International College of Rheumatology in 2015 [23]. Gouty arthritis cases were classified as SUA1-SUA4 according to SUA grading criteria [24] as follows: UA ≤ 4.2 mg/dl was classified as SUA1 level; 4.3 ≤ UA ≤ 5.2 mg/dl was classified as SUA2 level; 5.3 ≤ UA ≤ 6.3 mg/dl was classified as SUA3 level; and UA ≥ 6.4 mg/dl was classified as SUA4 level. The exclusion criteria were as follows: malignant tumors, hematologic diseases, hepatic or renal diseases, inflammatory diseases, basic psychoneurological diseases, and hyperthyroid diseases; recently consumed alcohol; taking drugs to promote (reduce) UA excretion (e.g., allopurinol and febuxostat benzbromarone); and taking drugs to relieve inflammation (steroidal anti-inflammatory drugs or non-steroidal anti-inflammatory drugs). Based on the examination results, all 890 healthy individuals had no history of inflammatory disease or relevant immune system disorders.

The following data were collected from all study subjects: basic personal information; personal behavior, such as smoking, alcohol consumption, and regular exercise; and general health indicators. Patients with gouty arthritis also provided detailed information about the

onset of gout. In addition, blood specimens were collected from all study subjects (gouty arthritis and health examinations). The collected blood specimens were used for the following tests: biochemical parameters; DNA extraction and single nucleotide polymorphism (SNP) detection; and functional assays for protein expression. This study was carried out from January 2016 and ended in December 2021, the questionnaire used during this period included informed consent from the participants, the participants included in this study were all adult males, there were no minors, informed consent was given at the time of inclusion of the participants and the participants were included in this study after giving their verbal consent, accept of questionnaire content and collection of blood specimens for the study. The experiments involved in this study have been recognized by the institution and have been strictly carried out in accordance with relevant guidelines and regulations.

## 2.2 Reagents and disposables

The reagents and disposables involved in this study mainly include SNP scanTM typing kit (SuZhou Tianhao biotechnology Co., China), TLR4 enzyme immunoassay kit (ELISA, China), Multiskan FC enzyme labeling instrument (Thermo Fisher Scientific, USA), 2720 Thermal Cycler (ABI, USA), 3130xl genetic analyze (ABI, USA), and so on.

## 2.3 Laboratory examination

Uric acid (UA), hyperuricemia (HUA), glucose (GLU), blood urea nitrogen (BUN), creatinine (CREA), plasma total cholesterol (TC), triglycerides (TG), high-density lipoprotein cholesterol (HDL-C), low-density lipoprotein cholesterol (LDL-C), creatinine clearance rate (Ccr). Apolipoprotein A (APOA), Apolipoprotein B (APOB), Lipoprotein a [Lp(a)], Systolic blood pressure (SBP), Diastolic blood pressure (DBP), Estimated glomerular filtration rate (eGFR). The measured indexes were completed in the hospital's laboratory department.

## 2.4 Whole blood DNA extraction testing

Thermo Scientific King Duo automated nucleic acid extractor (magnetic bead method) was used to extract DNA from collected whole blood, and 1 mL of DNA samples were extracted for quality testing and concentration estimation using agarose gel electrophoresis. By observing the number of bands and brightness of the Marker, the Molecular Weight Size and Concentration were compared to the DNA to be tested. In order to ensure the accuracy of the experimental results to reduce bias and error, the same type of automatic nucleic acid extractor and the same batch of DNA extraction kit were used to conduct the experiment, and the proposed DNA should be tested for DNA purity in a timely manner, and the samples with unqualified purity should be re-tested, and if the purity is still unqualified, then the samples should be discarded. Furthermore, all extracted DNA was kept frozen at -80˚C.

## 2.5 SNP selection in the *TLR4 gene* and primer synthesis

Based on the NCBI database (http://www.ncbi.nlm.nih.goc/), International Human Genome Haplotype Project (http://hapmap.ncbi.nlm.nih.gov/), SNP information, Ensembl database (http://asia.ensembel.org/index.html), and relevant domestic and international literature, the presence of SNPs in northern Chinese populations were identified. The screening of alleles in the Chinese population, combined with the results of previous studies and a large amount of literature suggests that there is an association between polymorphisms in the *TLR4 gene* and the development of gout. Taking into account the chromosome, physical location, and the proportion of genotypes in the Asian population, a total of 10 loci for the *TLR4 gene* were

**Table 1. Primer sequence information table.**

| SNP | 5'end probes1 | 5' end probes 2 | 3' end probes |
|---|---|---|---|
| rs2149356 | AAACATTAAGA GTATCTGTGACA CTTATGTGTACTG | AAACATTAAGA GTATCTGTGACA CTTATGTGTACTT | TTTCGTATCTCTG AAATTGATATTTACCAGTC |
| rs2737191 | GCAAACACTTCT AGGTCCCTGTCG | GCAAACACTTCT AGGTCCCTGTCA | AATATGGGATTC CTCCATTGACTGA |
| rs10759932 | GTTCTCATTTTTTCA CATCTTCACCATCG | GTTCTCATTTTTTCA CATCTTCACCACCA | CTTATTATTACTTTCT GGTCTGTGTAAAATGTGGTAAT |

screened, and three loci were selected for subsequent analyses. The Hardy-Weinberg balance test for SNP loci was performed with all control groups in this study, and all three loci in the control group complied with the Hardy-Weinberg balance test ($P > 0.05$), which can indicate that a state of genetic equilibrium has been reached in the Han population in this region.

Review of literature related to *TLR4 gene*, validation of the designed primers using Pubmed Blast method, matching primer sequences in the NCBI database, and the primers were synthesized by Shanghai Tianhao Biotechnology Co., Ltd. and stored frozen at -20°C in Table 1.

## 2.6 Assay for SNPs in the *TLR4 gene*

The SNPscan[TM] multiplex kit, a rapid multiplex point mutation detection technology, was used in all study subjects to type three loci of the *TLR4 gene* (1207 cases). Assay steps include sample lysis, ligation reaction pre-mix preparation, ligation reaction, and multiplex fluorescence PCR reaction. Finally, the results of *TLR4 gene* SNP typing are obtained in Fig 1.

## 2.7 Enzyme-linked immunosorbent assay (ELISA)

TLR4 protein expression levels in serum were measured in 41 gouty arthritis patients and 82 healthy people using a double antibody sandwich enzyme-linked immunosorbent assay (ELISA) kit. The absorbance value (OD) was measured at 450 nm using an enzyme standardizer. The color shade was proportional to the amount of target protein in the sample, and the amount of target protein in the sample was quantified by plotting the logarithm of optical density versus concentration and establishing a standard curve for the TLR4 protein of the sample to be calculated using a known concentration of TLR4 protein.

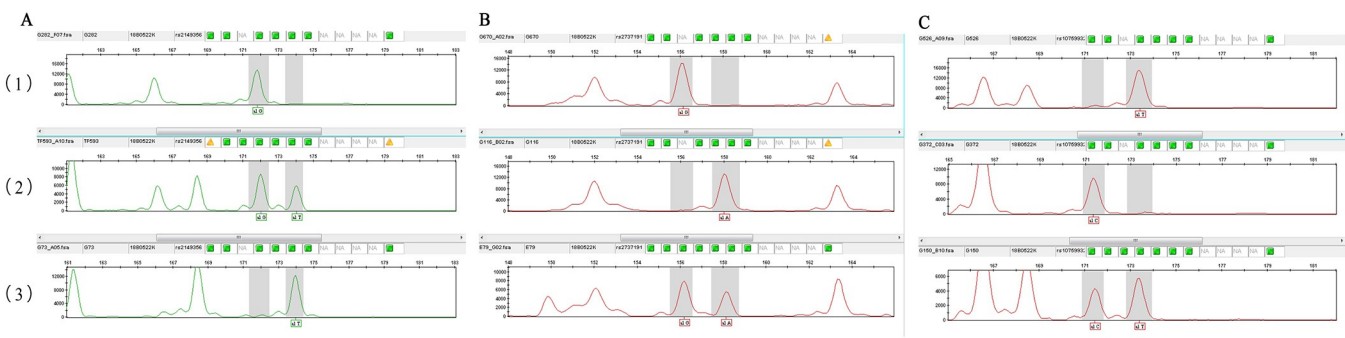

Figure A (1) is GG, (2) is GT, (3) is TT; Figure B (1) is GG, (2) is AA, (3) is GA; Figure C (1) is TT, (2) is CC, (3) is CT

**Fig 1.** A: Genotyping results of rs2149356, B: Genotyping results of rs2737191, C: Genotyping results of rs10759932.

## 2.8 Statistical analysis

The SNP Hardy-Weinberg equilibrium test was carried out on the study subjects using the Haploview software to ensure that the study subjects were chosen at random SPSS 26.0 was used for statistical analysis, the measurement data were first tested for normality, and if they obeyed a normal distribution, the mean ± standard deviation was used for statistical description, the analysis of variability between the two groups was performed using two independent samples t-test, one-way ANOVA was used for analysis of variance between multiple groups. If they did not obey normal distribution then non-parametric test was used for the analysis of differences between groups. Count data were expressed in % as the probability of distribution, and the χ2 test was used for the analysis of differences between groups. Pearson correlation coefficient for correlation analysis, Logistic regression model for analysis of factors influencing the development of gouty arthritis, and linear regression for correlation of gene-gene expression levels. Statistical significance was set at $P<0.05$. The two-tailed test was used for all analyses involving variability.

## 2.9 Ethics statements

Ethics approval This study was approved by the Ethics Committee of the First Affiliated Hospital of Xinjiang Medical University.

## 3. Results

### 3.1 Analysis of general clinical indicators

General clinical indicators were compared between gout and control groups in Table 2, and a normality test indicated that the general clinical indicators in this study conformed to normality. Age, body mass index (BMI), waist-to-hip ratio, UA, blood glucose, and diastolic blood pressure were all significantly higher in the gout group than in the control group. There was no significant difference in SBP between the two groups ($P<0.05$). The comparison of lipid and renal function indicators in the gout and control groups revealed that apolipoprotein B (APOB) and blood creatinine (CREA) levels were significantly higher in the gout group than in the control group. There was no significant difference in triglyceride (TG), total cholesterol (TC), high-density lipoprotein cholesterol (HDL-C), low-density lipoprotein cholesterol (LDL-C), apolipoprotein A (APOA), lipoprotein (a) [Lp(a)], blood urea nitrogen (BUN), creatinine clearance (Ccr), and estimated glomerular filtration rate (eGFR) levels between the two groups ($P>0.05$). The analysis of general behavioral indicators in the gout and control groups revealed that the composition ratios of alcohol consumption, regular exercise, obesity, hypertension, hyperglycemia, and kidney damage were significantly higher in the gout group than in the control group ($P<0.05$). There was no significant difference in the two groups smoking, hypertriglyceridemia, and hypercholesterolemia composition ratios.

### 3.2 Analysis of the clinical characteristics of the gout population

In gouty arthritis patients with varying UA levels, Table 3 shows the age of onset, causative factors, first joint, present cumulative joint, gout stone, and kidney stone were evaluated. The proportion of each index at the SUA1 level was low because the distribution of the SUA levels in gouty arthritis patients were generally greater than 4.75 mg/dl, and SUA grades were located at the SUA2-SUA4 levels with the SUA3 and SUA4 levels being predominant. Age showed a non-significant decreasing trend at the SUA2-SUA4 levels as UA levels increased ($P>0.05$). The overall number of triggering factors increased as the UA levels increased. Among these factors, the differences among alcohol consumption, high-fat diet, and physical exertion at different

**Table 2. Comparison of general indicators between gout and control group.**

| Variable | Control group(n = 890) | Gout group(n = 1207) | t/χ2 value | P-value |
|---|---|---|---|---|
| Age (years) | 45.97±13.94 | 47.96±12.45 | -2.366 | **0.018** |
| BMI (kg/m²) | 25.66±3.28 | 26.68±3.32 | -4.671 | **<0.001** |
| WHR | 0.91±0.05 | 0.95±0.06 | -8.214 | **<0.001** |
| SUA (µmol/L) | 404.24±93.33 | 511.82±131.01 | -15.735 | **<0.001** |
| GLU (mmol/L) | 5.44±1.50 | 5.84±1.89 | -3.387 | **0.001** |
| SBP (mmHg) | 124.99±13.62 | 125.48±13.79 | -0.543 | 0.587 |
| DBP (mmHg) | 77.45±9.67 | 81.16±11.63 | -5.544 | **<0.001** |
| TG(mmol/L) | 2.12±1.91 | 1.94±1.32 | 1.762 | 0.078 |
| TC(mmol/L) | 4.70±1.02 | 4.58±1.09 | 1.863 | 0.063 |
| HDL-C(mmol/L) | 1.29±0.29 | 1.41±0.84 | -0.370 | 0.711 |
| LDL-C(mmol/L) | 2.70±0.83 | 2.69±0.78 | 0.116 | 0.907 |
| APOA(g/L) | 1.21±0.26 | 1.96±0.93 | -1.231 | 0.220A |
| APOB(g/L) | 0.89±0.28 | 1.01±0.34 | -3.590 | **<0.001** |
| Lp(a) (mg/L) | 185.65±63.91 | 176.13±61.12 | 0.508 | 0.612 |
| BUN(mmol/L) | 5.38±1.61 | 5.47±2.04 | -0.644 | 0.520 |
| CREA(µmol/L) | 87.95±17.36 | 92.76±28.73 | -2.803 | **0.005** |
| Ccr(mL/min) | 103.65±29.65 | 106.55±39.87 | -1.181 | 0.238 |
| eGFR(mL/min) | 93.43±20.19 | 93.49±35.76 | -0.032 | 0.975 |
| Smoke | 371(41.7) | 116(36.6) | 2.518 | 0.113 |
| drink | 331(37.3) | 153(48.4) | 11.831 | **0.001** |
| exercise | 281(31.6) | 44(13.9) | 37.188 | **<0.001** |
| obesity | 184(20.7) | 85(27.8) | 6.591 | **0.010** |
| hypertension | 440(49.4) | 222(70.0) | 40.027 | **<0.001** |
| hyperglycemia | 135(15.2) | 93(29.3) | 30.628 | **<0.001** |
| hypertriglyceridemia | 246(27.6) | 96(30.3) | 0.804 | 0.370 |
| hypercholesteremia | 52(5.8) | 19(6.0) | 0.010 | 0.922 |
| Renal insufficiency | 393(44.5) | 171(53.9) | 8.430 | **0.004** |

BMI: Body mass index, WHR: Waist-to-hip ratio, SUA: Serum uric acid, GLU: Glucose, SBP: Systolic blood pressure, DBP: Diastolic blood pressure, TG: Triglyceride, TC: Total cholesterol, HDL-C: High-density lipoprotein cholesterol, LDL-C: Low-density lipoprotein cholesterol, APOA: Apolipoprotein A, APOB: Apolipoprotein B, Lp(a): Lipoprotein (a), BUN: Blood urea nitrogen, CREA: Creatinine, Ccr: Creatinine clearance, eGFR: Estimated glomerular filtration rate.

UA levels were statistically significant ($P<0.05$), and there were no statistically significant differences between seafood and cold exposure ($P>0.05$). Regarding the first site of gouty arthritis, the highest percentage was in the first toe knuckle, and as UA levels increased, the percentage of first sites increased with the overall number concentrated in the SUA4 group. At different UA levels, there were statistically significant differences in the first toe knuckle, dorsal foot, ankle, and knee joints ($P<0.05$). In the other toe and finger joints, Achilles tendon, elbow joint, wrist joint, and under the shoulder joint, no statistically significant differences were found ($P>0.05$). The proportion of accumulation of the first toe knuckle and ankle joints was predominant in terms of the current accumulation site, and the overall number was concentrated in the SUA4 group. At different UA levels, there was a statistically significant difference in the number of gout stones ($P<0.05$), but there was no statistically significant difference in the number of kidney stones ($P>0.05$).

The results of the analysis using a one-way logistic regression analysis model showed that age, alcohol consumption, regular exercise, obesity, high uric acid, high glucose, renal insufficiency and hypertension were associated with the development of gout ($P<0.05$). In the multi-

**Table 3. Comparison of clinical characteristics of gout with different uric acid levels.**

| Variable n(%) | | SUA1 | SUA2 | SUA3 | SUA4 | P-value |
|---|---|---|---|---|---|---|
| onset age years (SD) | | 46.71 (5.77) | 45.64 (17.09) | 45.86 (11.36) | 40.67 (11.68) | 0.077[#] |
| predisposition | seafood | 0(0.0) | 1(0.8) | 2(7.7) | 23(88.5) | 0.089[*] |
| | beer | 2(2.6) | 3(3.8) | 4(5.1) | 69(88.5) | **<0.001** |
| | high-fat | 1(1.2) | 3(3.6) | 6(7.1) | 74(88.1) | **<0.001** |
| | cold | 0(0.0) | 1(6.7) | 1(6.7) | 13(86.7) | 0.349[*] |
| | tired | 2(7.1) | 0(0.0) | 1(3.6) | 25(89.3) | **0.009**[*] |
| | emotion | 0(0.0) | 0(0.0) | 0(0.0) | 2(100.0) | 0.783[*] |
| First attack | first digit | 5(2.2) | 8(3.5) | 16(7.1) | 197(87.2) | **<0.001** |
| | other digit | 1(9.1) | 1(9.1) | 2(18.2) | 7(63.6) | 0.689[*] |
| | heel tendon | 0(0.0) | 1(16.7) | 0(0.0) | 5(83.3) | 0.585[*] |
| | dorsum pedis | 0(0.0) | 0(0.0) | 1(3.7) | 26(96.3) | **0.008**[*] |
| | ankle joint | 2(2.9) | 1(1.5) | 3(4.4) | 62(91.2) | **<0.001** |
| | knee joint | 0(0.0) | 1(2.9) | 1(2.9) | 32(94.1) | **0.004**[*] |
| | elbow joint | 0(0.0) | 0(0.0) | 1(14.3) | 6(85.7) | 0.660[*] |
| | wrist joint | 0(0.0) | 1(11.1) | 2(22.2) | 6(66.7) | 0.964[*] |
| | shoulder joint | 1(25.0) | 1(25.0) | 1(25.0) | 1(25.0) | 0.051[*] |
| current attack | first digit | 5(2.3) | 9(4.1) | 17(7.7) | 189(85.9) | **<0.001** |
| | other digit | 1(5.0) | 1(5.0) | 3(15.0) | 15(75.0) | 0.660[*] |
| | heel tendon | 2(6.9) | 3(10.3) | 6(20.7) | 18(62.1) | 0.664 |
| | dorsum pedis | 1(1.9) | 2(3.7) | 1(1.9) | 50(92.6) | **<0.001** |
| | ankle joint | 2(1.7) | 3(2.5) | 8(6.6) | 108(89.3) | **<0.001** |
| | knee joint | 4(3.8) | 4(3.8) | 8(7.7) | 88(84.6) | **<0.001** |
| | elbow joint | 0(0.0) | 0(0.0) | 1(3.4) | 28(96.6) | **0.005**[*] |
| | wrist joint | 1(3.0) | 1(3.0) | 2(6.1) | 29(87.9) | **0.034**[*] |
| | shoulder joint | 1(5.9) | 2(11.8) | 2(11.8) | 12(70.6) | 0.733[*] |
| gout stone | | 0(0.0) | 1(3.0) | 0(0.0) | 32(97.0) | **0.001**[*] |
| renal calculus | | 1(5.6) | 0(0.0) | 3(16.7) | 14(77.8) | 0.388 |

[#]One-way ANOVA

[*]Fisher's exact probability.

factor logistic regression analysis model included in the multi-factor logistic regression analysis model in Table 4, alcohol consumption, high uric acid, high blood sugar, and high blood pressure were found to be risk factors for the development of gout (*P<0.05, OR>1*). regular exercise is a protective factor in the development of gout (*P<0.05, OR<1*).

### 3.3 Differential analysis of *TLR4 gene* SNPs and protein expression

In the codominant model Table 5 shows the AG genotype of the *TLR4 gene* rs2737191 polymorphism increased the risk of gout compared to the AA genotype (*OR = 2.249, 95%CI 1.010~5.008*), while the G allele increased the risk of gout compared to the A allele (*OR = 2.227, 95%CI 1.006~4.932*). In the codominant, dominant, and recessive models, the rs2149356 and rs10759932 loci were no statistically significant difference between gout and control groups (*P>0.05*).

Serum samples were collected from 41 gouty arthritis patients (gout group) and 82 healthy individuals (control group), and TLR4 protein concentrations were determined. In Fig 2A the results revealed that the TLR4 protein concentration in the gout group (19.19±3.09 ng/ml) was

**Table 4. Multiple factor logistic regression analysis of gout incidence.**

| Variable | β | S.E. | Waldχ2 | P-value | OR (95%CI) |
|---|---|---|---|---|---|
| constant | -3.631 | 0.390 | 86.546 | <0.001 | - |
| Age | 0.002 | 0.006 | 0.092 | 0.762 | 1.002(0.990~1.013) |
| Drink | 0.647 | 0.151 | 18.437 | <0.001 | 1.911(1.422~2.568) |
| Exercise | -0.637 | 0.194 | 10.799 | 0.001 | 0.529(0.362~0.773) |
| Obesity | 0.009 | 0.169 | 0.003 | 0.957 | 1.009(0.724~1.406) |
| Hyperuricemia | 1.557 | 0.172 | 81.952 | <0.001 | 4.744(3.386~6.645) |
| Hyperglycemia | 0.605 | 0.176 | 11.812 | 0.001 | 1.832(1.297~2.587) |
| Renal insufficiency | 0.184 | 0.153 | 1.445 | 0.229 | 1.203(0.890~1.625) |
| Hypertension | 0.650 | 0.154 | 17.744 | <0.001 | 1.916(1.416~2.593) |

significantly higher than that in the control group (15.85±4.75 ng/ml). The difference in protein expression levels between the two groups was statistically significant ($t$ = -4.967, $P<0.001$).

The UA levels of all study subjects were graded, and Table 6 shows the results of one-way analysis of variance (ANOVA) revealed statistically significant differences in TLR4 protein expression levels at different UA levels ($P<0.05$). A two-by-two least significant difference (LSD) comparison was performed, and the results revealed statistically significant differences in TLR4 protein expression in the following comparisons: between SUA1 and SUA3; between SUA1 and SUA4; between SUA2 and SUA3; and between SUA2 and SUA4 ($P<0.05$) in Fig 2B. In the gout group, a stratified analysis of disease duration, gout stone kidney stone status, and smoking and alcohol consumption was performed. It was found that the differences in TLR4 protein expression between different subgroups of the above factors were not statistically significant ($P<0.05$).

The association of each index with TLR4 protein expression was investigated using bivariate correlation analysis in Table 7, which demonstrated that the concentration of TLR4 protein expression was related only to UA levels and no other indicators.

Comparison of protein expression levels between different *TLR4 gene* polymorphisms in the gout and control groups using a codominant model indicated that TLR4 protein expression levels differed between the following comparisons in Table 8: the GG and GT genotypes of the rs2149356 polymorphism ($P<0.05$); the AA and AG genotypes of the rs2737191 polymorphism ($P<0.05$); and the TT and TC genotypes of the rs10759932 polymorphism ($P<0.05$).

## 4. Discussion

Gouty arthritis is currently a relatively common metabolic disease, and the incidence of gouty arthritis is increasing year by year due to changes in lifestyle and dietary habits [25]. The mean age of onset of gouty arthritis in patients included in the present study was 47.96 ± 12.45 years compared to the mean age of onset of 48.28 years previously reported in a similar study in China, confirming a younger age of onset of gouty arthritis [26]. Urate crystals are deposited in high concentrations in the joints and surrounding soft tissues with the first joints primarily being the toe and finger joints followed by ankle joints, knee joints, back of the foot, finger joints, and elbow joints [27,28]. When gout stones form, they seriously affect the patient's life and reduce the quality of life [29]. The inflammatory burst of hyperuricemia is the main cause of gouty arthritis. The present study showed that the UA level in the gout group was significantly higher than that in the control group, and multifactorial logistic regression analysis confirmed that hyperuricemia was a risk factor for gouty arthritis ($OR = 4.744$). In a prospective

**Table 5. Different genetic models of three loci and gout susceptibility.**

| SNP | Genetic model | Control (n = 890) | Gout (n = 1207) | P-value | OR (95%CI) |
|---|---|---|---|---|---|
| rs2149356 | codominant | | | | |
| | GG | 303(34.0) | 104(32.8) | 0.624 | 1 |
| | GT | 434(48.8) | 164(51.7) | | 1.101(0.827~1.446) |
| | TT | 153(17.2) | 49(15.5) | | 0.966(0.794~1.175) |
| | dominant | | | | |
| | GG | 303(34.0) | 104(32.8) | 0.689 | 1 |
| | GT+TT | 587(66.0) | 213(67.2) | | 1.057(0.805~1.388) |
| | recessive | | | | |
| | GG+GT | 737(82.8) | 268(84.5) | 0.478 | 1 |
| | TT | 153(17.2) | 49(15.5) | | 0.881(0.620~1.251) |
| | Allele | | | | |
| | G | 1040(58.4) | 372(58.7) | 0.913 | 1 |
| | T | 740(41.6) | 262(41.3) | | 0.990(0.823~1.190) |
| rs2737191 | codominant | | | | |
| | AA | 876(98.4) | 306(96.5) | **0.042** | 1 |
| | AG | 14(1.6) | 11(3.5) | | 2.249(1.010~5.008) |
| | Allele | | | | |
| | A | 1766(99.2) | 623(98.3) | **0.043** | 1 |
| | G | 14(0.8) | 11(1.7) | | 2.227(1.006~4.932) |
| rs10759932 | codominant | | | | |
| | TT | 448(50.3) | 141(44.5) | 0.200 | 1 |
| | TC | 365(41.0) | 146(46.1) | | 1.271(0.971~1.664) |
| | CC | 77(8.7) | 30(9.4) | | 1.113(0.883~1.402) |
| | dominant | | | | |
| | TT | 448(50.3) | 141(44.5) | 0.073 | 1 |
| | TC+CC | 442(49.7) | 176(55.5) | | 1.265(0.978~1.637) |
| | recessive | | | | |
| | TT+TC | 813(91.3) | 287(90.5) | 0.662 | 1 |
| | CC | 77(8.7) | 30(9.5) | | 1.104(0.709~1.718) |
| | Allele | | | | |
| | T | 1261(70.8) | 428(17.7) | 0.116 | 1 |
| | C | 519(29.2) | 206(32.5) | | 1.169(0.962~1.421) |

study, Lin et al. [30] showed that the 5-year cumulative incidence and SUA level show a positive correlation. Excess UA in the body is primarily caused by consuming large amounts of high purine foods, such as seafood, alcohol, and high-fat foods, which are closely related to the body's glycolipid functions. Furthermore, studies have shown that high UA levels are linked to type II diabetes and hypertension [31–34]. This is closely related to the function of the kidneys, which excrete UA in the body [35–37], and long-term high blood sugar and high blood pressure cause kidney dysfunction. In the kidney, UA is first filtered by the glomerulus and then reabsorbed, and it is secreted by the renal tubules and finally excreted in the urine. Therefore, when the ability of the kidneys to excrete UA is blocked, UA levels in the body are increased, and the long-term high UA level in the body increases the occurrence of gouty arthritis. The present findings supported the associations between gout and the development of hypertension, hyperglycemia, and renal function.

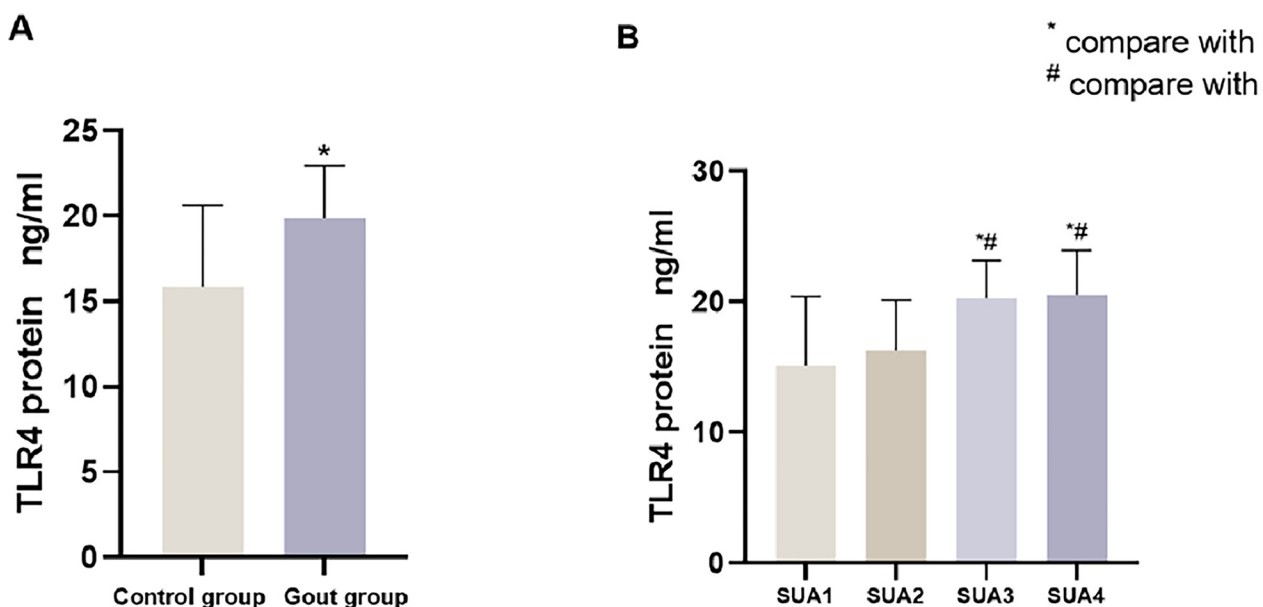

**Fig 2.** A: Comparison of TLR4 protein concentration between gout group and control group, B: Comparison of TLR4 protein concentration between different SUA grades.

Gouty arthritis occurrence is also linked to genetic factors. The current pathogenesis of gouty arthritis is primarily based on the following two main processes: activated NALP3 inflammasome assembly mechanism and the TLR2/4-NF-κB signal transduction pathway. TLRs are the most important receptors in the inflammatory pathway, and they are expressed in a wide range of immune cells, recognizing the pathogen-associated molecular patterns that invade the organism as well as danger-associated molecular patterns. UA salt is a danger-associated signal that activates the TLR inflammatory pathway and secretion of various inflammatory factors [such as interleukin (IL)-1 and IL-8], which initiates the immune response, resulting in activation of the inflammatory pathway [38]. The *TLR4 gene* is located on human chromosome 9q32-9q33. In animal experiments, researchers confirmed the role of the *TLR4 gene* in the urate-mediated inflammatory cascade response in gout [39]. Traditional Chinese Medicine studies have shown that TLR4 is overexpressed in a gout mouse model and that anti-inflammatory treatment reduces TLR4 protein expression [40]. Research has shown that rs10759932 polymorphisms in the TLR4 are associated with different inflammatory diseases, such as chronic obstructive pneumonia, tuberculosis, and other diseases [41]. Moreover, childhood asthma is linked to rs2737191 polymorphisms [42], and rs2149356 polymorphisms have been linked to gout in Sichuan, China [43].

Gouty arthritis is a complex polygenic disease caused by a combination of genetic and environmental factors. Researchers now believe that a variety of risk factors for gout exist, including hyperuricemia, dyslipidemia, kidney damage, and obesity, and all of which are linked to the development of gouty arthritis. In the male population, individuals with the AA genotype of the *TLR4 gene* rs2737191 polymorphism had a significantly lower risk of developing gouty arthritis than individuals with G alleles, and the rs2149356 and rs10759932 polymorphisms were not significantly associated with the development of gouty arthritis in the male gouty arthritis population. In addition, TLR4 protein was highly expressed in the gouty arthritis population, and the concentration of TLR4 protein increased as the UA levels increased in this population. These findings suggested that the AG genotype of the *TLR4 gene* rs2737191

**Table 6. Expression of TLR4 protein in different course and uric acid grade.**

| Variable | TLR4(ng/ml) | F- value | P-value |
|---|---|---|---|
| SUA classification | | | |
| SUA1 | 15.15±5.26 | 11.502 | **0.001** |
| SUA2 | 16.25±3.85 | | |
| SUA3 | 20.27±2.90[*#] | | |
| SUA4 | 20.53±3.42[*#] | | |
| Gout course(n = 41) | | | |
| 1–5 years | 20.29±2.98 | 0.481 | 0.622 |
| 6-10yesrs | 20.09±3.45 | | |
| >11years | 19.13±2.97 | | |
| Gout stone(n = 41) | | | |
| Yes | 20.13±3.75 | 0.203 | 0.840 |
| No | 19.87±3.01 | | |
| Renal calculus(n = 41) | | | |
| Yes | 19.87±2.29 | -0.030 | 0.977 |
| No | 19.92±3.19 | | |
| Exercise(n = 41) | | | |
| Yes | 21.48±3.22 | -1.367 | 0.180 |
| No | 19.64±3.04 | | |
| Smoke(n = 41) | | | |
| Yes | 20.37±3.53 | 0.925 | 0.361 |
| No | 19.48±2.62 | | |
| Drink(n = 41) | | | |
| Yes | 20.63±3.33 | 1.710 | 0.095 |
| No | 19.00±2.56 | | |

[*] Comparison with SUA1 group(P<0.001)

[#] comparison with SUA2 group(P<0.001).

**Table 7. Correlation between TLR4 protein expression and biochemical indicators.**

| Variable | Control Group(n = 890) | | Gout Group(n = 1207) | |
|---|---|---|---|---|
| | R-value | P-value | R-value | P-value |
| Age(years) | -0.032 | 0.777 | -0.005 | 0.975 |
| SUA(µmol/L) | 0.040 | 0.803 | 0.442 | <**0.001** |
| GLU(mmol/L) | 0.217 | 0.050 | -0.098 | 0.541 |
| BUN(mmol/L) | -0.047 | 0.672 | -0.189 | -0.237 |
| CREA(µmol/L) | 0.047 | 0.677 | 0.114 | 0.477 |
| eGFR(µmol/L) | 0.007 | 0.947 | -0.041 | 0.797 |
| Ccr(mL/min) | 0.097 | 0.387 | 0.069 | 0.671 |
| TG(mmol/L) | 0.049 | 0.661 | 0.232 | 0.144 |
| TC(mmol/L) | -0.192 | 0.085 | 0.082 | 0.611 |
| HDL-C(mmol/L) | -0.131 | 0.240 | -0.059 | 0.714 |
| LDL-C(mmol/L) | -0.116 | 0.299 | 0.023 | 0.888 |
| BMI(kg/m$^2$) | 0.057 | 0.611 | 0.182 | 0.262 |

SUA: Serum uric acid, GLU: Glucose, BUN: Blood urea nitrogen, CREA: Creatinine, Ccr: Creatinine clearance, TG: Triglyceride, TC: Total cholesterol, HDL-C: High-density lipoprotein cholesterol, LDL-C: Low-density lipoprotein cholesterol, BMI: Body mass index.

**Table 8. Comparison of TLR4 protein concentration of different SNPs.**

| SNP | Control(n = 890) | Gout(n = 1207) | t- value | P-value |
|---|---|---|---|---|
| *rs2149356* | | | | |
| GG | 15.61±5.02 | 19.19±2.49 | -2.807 | **0.007** |
| GT | 15.98±4.89 | 20.49±3.52 | -3.746 | **<0.001** |
| TT | 15.97±4.29 | 20.28±2.56 | -0.979 | 0.341 |
| *rs2737191* | | | | |
| AA | 15.92±4.82 | 19.80±3.09 | -4.534 | **<0.001** |
| AG | 14.18±1.97 | 21.34±3.32 | -3.211 | **0.033** |
| *rs10759932* | | | | |
| TT | 15.99±5.17 | 19.41±2.68 | -2.767 | **0.008** |
| TC | 15.41±4.41 | 20.40±3.52 | -4.315 | **<0.001** |
| CC | 17.23±3.92 | 20.28±2.15 | -0.729 | 0.494 |

polymorphism is a risk factor for the development of gout in Han Chinese males in northwest China and that rs2737191 polymorphisms lead to high TLR4 protein expression, resulting in increased risk of developing gouty arthritis. Moreover, this increase may also be related to the involvement of *TLR4 genes* in the regulation of immune, inflammatory, and lipid metabolic processes [43]. Although the important role of the TLR4-NF-κB signaling pathway in the pathogenesis of gouty arthritis has been demonstrated, few drugs have been studied to target this pathway for the treatment of gouty arthritis. To explore the target drug from the perspective of "TLR4—NFκB signaling pathway" to inhibit the level of TLR4 protein, which may provide a new therapeutic idea to relieve the symptoms of gouty arthritis in the future. Furthermore, there have been few reports on gene polymorphisms and protein expression differences among the three polymorphisms of the *TLR4 gene* selected for this study. The present study demonstrated that these polymorphisms differ between gouty arthritis patients and healthy individuals, thereby contributing to the further investigation of the etiology and treatment of gouty arthritis from a genetic perspective.

## 5. Limitations

The population in the present study was limited to one region, indicating that the single nucleotide differences and protein expression may not be applicable to other regions. Further in-depth research is required to broaden the research population selection and add interaction to validate the present conclusions. In addition, more research is needed to increase the sample size for undetected pure mutants, such as the GG genotype of the rs2737191 polymorphism.

## 6. Future perspective

Currently, it is believed that gouty arthritis is induced by the release of a large number of inflammatory cytokines, including IL-1β, IL-6, tumor necrosis factor-alpha (TNF-α) and other related inflammatory cytokines, in addition to those investigated in the present study. As a result, additional research into the genes of inflammatory signaling pathways and related inflammatory cytokines is required to discover important pathogenic factors affecting gout, to investigate environmental factors, to investigate multifactorial and multilevel pathogenic factors affecting the development of gout at the genetic and environmental levels, and to provide a foundation for future gout control and targeted treatment of gout.

## 7. Conclusions

The *TLR4 gene* rs2737191 polymorphism is associated with gout susceptibility, whereas the rs2149356 and rs10759932 polymorphisms are not significantly associated with gout occurrence. Moreover, TLR4 protein is highly expressed in the gout population, and it increases with increasing UA levels in this population. The use of genetic differences as markers for gout screening in high-risk populations is expected to be an important diagnostic tool in the future. Based on the findings of the environmental factor investigation, gout patients or individuals at risk are encouraged to exercise because regular exercise helps to boost immunity, protect joint function, and avoid risk factors for the occurrence of gout. In addition, these individuals are encouraged to lower alcohol consumption, control body weight, protect kidney function, and lower the occurrence of glucose and lipid metabolism disorders, which may reduce the occurrence of gout to some extent.

## Acknowledgments

## Declarations

Consent to participate All subjects signed the informed consent. Consent to publish The authors affirm that human research participants provided informed consent for publication. The datasets generated and/or analysed during the current study are available in the NCBI repository. Submitter batch id: 18B0522K-1~-3 (https://www.ncbi.nlm.nih.gov/SNP/snp_viewTable.cgi?handle=XJMUCL).

## Author Contributions

**Conceptualization:** Xiaobin Guo, Lei Miao.

**Data curation:** Lu Liu, Shuang He, Hua Yao, Dan Zhou, Xiaobin Guo.

**Formal analysis:** Lu Liu, Xiaobin Guo.

**Funding acquisition:** Lei Miao.

**Investigation:** Lu Liu, Shuang He.

**Methodology:** Lu Liu, Xiaobin Guo, Lei Miao.

**Supervision:** Hua Yao, Xiaobin Guo, Lei Miao.

**Writing – original draft:** Lu Liu, Shuang He, Lin Jia.

**Writing – review & editing:** Lu Liu, Shuang He.

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
