## [Decision Letter · Decision Letter 0]

3 Jan 2024

PONE-D-23-37483Correlation analysis of serum TLR4 protein levels and TLR4 gene  polymorphisms in gouty arthritis patientsPLOS ONE

Dear Dr. Miao,

Thank you for submitting your manuscript to PLOS ONE. After careful consideration, we feel that it has merit but does not fully meet PLOS ONE’s publication criteria as it currently stands. Therefore, we invite you to submit a revised version of the manuscript that addresses the points raised during the review process.

We look forward to receiving your revised manuscript.

Kind regards,

Philippe T. Georgel

Academic Editor

PLOS ONE

Journal Requirements:

Natural Science Foundation of Xinjiang(2022D01C756,2020D01C263), Special fund project of youth scientific Research sailing from The First Affiliated Hospital of Xinjiang Medical University (2022-74), 

Natural Science Foundation of China (81460153, 82002276).

We are grateful for the support from the Natural Science Foundation of Xinjiang(2022D01C756,2020D01C263), Special fund project of youth scientific Research sailing from The First Affiliated Hospital of Xinjiang Medical University (2022-74), Natural Science Foundation of China (81460153, 82002276).

Natural Science Foundation of Xinjiang(2022D01C756,2020D01C263), Special fund project of youth scientific Research sailing from The First Affiliated Hospital of Xinjiang Medical University (2022-74), 

Natural Science Foundation of China (81460153, 82002276).

Additional Editor Comments:

Dear Dr. Lei Miao,

After careful consideration and consultation of the comments provided by the reviewers, there are several issues which would need to be addressed for your submitted material to become acceptable for publication by PLOS ONE. The main issues are outlined by the first reviewer and are mostly related to the lack of information related to TLR functions related to gout, and perhaps more importantly, the choice of statistical tools selected for the analysis of the results.

The discussion section would require addressing a more general conclusion, expanding outside of the scope of the tested sub-population.

If you decide to submit a revised version of the manuscript, please make sure that you address or answer all the various comments outlined by the reviewers.

Sincerely,

Philippe T. Georgel, PhD

Academic Editor for PLOS ONE

Reviewers' comments:

Reviewer's Responses to Questions

**Comments to the Author**

1. Is the manuscript technically sound, and do the data support the conclusions?

Reviewer #1: Partly

Reviewer #2: Yes

2. Has the statistical analysis been performed appropriately and rigorously? 

Reviewer #1: I Don't Know

Reviewer #2: Yes

3. Have the authors made all data underlying the findings in their manuscript fully available?

Reviewer #1: Yes

Reviewer #2: Yes

4. Is the manuscript presented in an intelligible fashion and written in standard English?

Reviewer #1: Yes

Reviewer #2: Yes

5. Review Comments to the Author

Reviewer #1: In this paper, Liu et al set out to identify possible correlations between TLR4 polymorphisms and plasma concentration and gouty arthritis. They did so by recruiting a 1207 study partipants. Even though their conclusions do appear sound, the work has some limitations that require improvement.

1-Overall this manuscript is very sparse on properly referring to previous work. This is most notorious in the introduction section. References ought to be added to the following sentences:

a) TLR expression by immune cells have been throuroughly described by multiple research papers and review articles. In this way, the sentence “TLR is the upstream signal of NF-κB in the TLR inflammatory pathway, which is an important pattern recognition receptor in the immune system, and it is present in various immune cells (T cells, B cells, and natural killer cells) as well as in monocytes, macrophages, neutrophils, and epithelial cells.” Ought to contain at least a reference to a review article such as, Fitzgerald KA, Cell 2020

b) Multiple papers have linked TLRs, and particularly TLR4, to immunological diseases in humans. Thus, the sentence “TLR-related genes have been linked to immunological illnesses, but little is known about their etiology and the factors that influence gout.” ought to be complemented by the following references:

- Reynolds, J. M Proc. Natl Acad. Sci. 2012.

-Amaral-Silva, D. Commun. Biol. 2021

- Ospelt, C. Arthritis Rheum. 2008

c) Even more pertinently to the work being presented in this manuscript, it misses the references where urate crystals have been identified as TLRs ligands in the sentence “For example, in patients with gouty arthritis, the release of urate crystals from damaged cells is an endogenous danger signal, thus indicating a link between gout pathogenesis and TLRs.”

2-Figure/Table legends ought to specify the number of study participants for each group that were sampled.

3- Similarly, the statistical tests used for each figure and table must be specified in figure/table legends

4-In the discussion section it is not clear what the authors mean by “non-specific immune cells” This denomination should be either clarified or removed from the text.

Reviewer #2: 1. What were the specific requirements for the quality and concentration of sample DNA in this study?

2. What was the basis for selecting rs2149356, rs2737191and rs10759932 loci of TLR4 gene in this study?

3. All data in Table 1 could be directly viewed in the database without any information added in this study. Was a separate list appropriate?

4. Please added the manufacturers and product numbers of the main kits in this study.

5. Why were the genotyping results not validated by sampling sequencing?

6. Why did the author not choose a two-tailed test during statistical analysis?

7. Table 3, 4 and 5 were the basic characteristics of the research objects, and it was suggested to combine the three tables.

8. Did the three loci of the TLR4 gene in this study conform to Hardy-Weinberg equilibrium?

9. The research object of this article is limited to a population in one region, and the conclusion does not emphasize whether the specific population is appropriate?

6. PLOS authors have the option to publish the peer review history of their article (what does this mean?). If published, this will include your full peer review and any attached files.

Reviewer #1: No

Reviewer #2: No

---

## [Author Response · Author response to Decision Letter 0]

18 Jan 2024

Thank you for your letter and for reviewer’ comments concerning our manuscript entitled “Correlation analysis of serum TLR4 protein levels and TLR4 gene polymorphisms in gouty arthritis patients”. These comments are all valuable and very helpful for revising and improving our paper, as well as the important guiding significance to our researches. We have studied comments carefully and have made corrections which we hope meet with approval. Our response is given in normal font and changes to the manuscript are given in the rad text. The responds to the comments are as follows:

1.With regard to "provide additional details regarding participant consent", informed consent of the participants in this study has been added to the methodology in the text as follows: The participants included in this study were all adult males, and there were no minors, who were informed of the informed consent at the time of inclusion. The participants included in this study were all adult males, with no minors, and were informed of the informed consent form at the time of inclusion, and were included in the study after verbal consent was given by the participants.

2.Role of Funder statement: Natural Science Foundation of Xinjiang(2022D01C756) recipient is Lu Liu, Natural Science Foundation of Xinjiang(2020D01C263) and Natural Science Foundation of China (82002276) recipient is Xiaobin Guo, Natural Science Foundation of China (81460153) recipient is Lei Miao. All three of the above grantees participated in this study, and their contributions are described in the Author Contributions section, and they are all credited in this manuscript.

3.According to the statement of financial support for manuscripts that have been deleted at your request.

4.The data availability statement you mentioned, all the authors of this study agree to data sharing, we will share the data according to your request, but we would like to set up access restrictions to our research data before acceptance of the paper. 

Response to Reviewer #1's comments:

1.Regarding your suggestion that some references to the results of previous generations are missing in the introduction, we have added references and labeled them in the revised draft.

2.The suggestion that the legends in the figures and tables should indicate the number of study participants sampled in each group has been modified and is indicated in the revised draft.

3.Regarding "The statistical tests used in each figure and table must be described in the legend of the figure/table", we have revised and labeled the figures and tables in the manuscript if they need to be revised. 

4.Regarding your suggestion that in the discussion section it is not clear what the authors mean by “non-specific immune cells”, this denomination has been removed from the manuscript.

Response to Reviewer #2's comments:

1.The question "What are the specific requirements for the quality and concentration of DNA in the samples for this study?" has been modified and added to the manuscript in section 2.4 Whole blood DNA extraction testing. We have modified this statement and added it to the "2.4 Whole blood DNA extraction testing" section of the manuscript.

2.Regarding "What was the basis for the selection of the rs2149356, rs2737191 and rs10759932 loci of the TLR4 gene in this study?" We have supplemented the manuscript with 2.5 SNP selection in the TLR4 gene and primer synthesis section.

3.In response to the question "Is it appropriate to have a separate list of information in Table 1?", we have deleted Table 1 and labeled it in the revised version.

4.Regarding "Please add the manufacturer and product numbers of the primary kits used in this study." We have added this to the "2.2 Reagents and disposables" section of the manuscript.

5.Why were the genotyping results not verified by sampling and sequencing? The genotyping in this study was not validated by sequencing, but the genotyping of the three loci studied in this paper was performed using the Multiple Point Mutation Rapid Test, which has the advantage of being easy to perform and highly sensitive.

6.Regarding "Why did the authors not choose two-tailed tests in their statistical analysis?", the two-tailed test was used for all analyses involving variability in this study, and has been added to the "2.8 Statistical analysis" section.

7.We have combined Tables 3, 4, and 5 into one table in the Table 2 section of the revised draft and have highlighted them in red.

8.Regarding "Do the three loci of the TLR4 gene in this study fulfill Hardy-Weinberg equilibrium?" The present study fulfills the Hardy-Weinberg equilibrium test and is supplemented in "2.5 SNP selection in the TLR4 gene and primer synthesis" in the manuscript.

9.Regarding the question "Is it appropriate that the object of study in this paper is limited to the population of one region and the conclusion does not emphasize a specific population?" The object of this study are coming to a region, there may be poor population extrapolation, but the authors of this paper this study still has its significance, the region's population distribution is not limited to one ethnic group, is a multi-region, multi-ethnic inclusion of the provinces and cities, although the investigation of the data is a region, but there is no China's population distribution there are large differences.

---

## [Decision Letter · Decision Letter 1]

1 Mar 2024

Correlation analysis of serum TLR4 protein levels and TLR4 gene  polymorphisms in gouty arthritis patients

PONE-D-23-37483R1

Dear Dr. Lei Miao,

We’re pleased to inform you that your manuscript has been judged scientifically suitable for publication and will be formally accepted for publication once it meets all outstanding technical requirements.

Kind regards,

Philippe T. Georgel

Academic Editor

PLOS ONE

Additional Editor Comments (optional):

Reviewers' comments:

Reviewer's Responses to Questions

**Comments to the Author**

1. If the authors have adequately addressed your comments raised in a previous round of review and you feel that this manuscript is now acceptable for publication, you may indicate that here to bypass the “Comments to the Author” section, enter your conflict of interest statement in the “Confidential to Editor” section, and submit your "Accept" recommendation.

Reviewer #1: All comments have been addressed

Reviewer #2: All comments have been addressed

2. Is the manuscript technically sound, and do the data support the conclusions?

Reviewer #1: Yes

Reviewer #2: Yes

3. Has the statistical analysis been performed appropriately and rigorously? 

Reviewer #1: Yes

Reviewer #2: Yes

4. Have the authors made all data underlying the findings in their manuscript fully available?

Reviewer #1: Yes

Reviewer #2: Yes

5. Is the manuscript presented in an intelligible fashion and written in standard English?

Reviewer #1: Yes

Reviewer #2: Yes

6. Review Comments to the Author

Reviewer #1: All my initial revision requests were satisfactorily addressed. At this time I have no more comments to make

Reviewer #2: The author has made appropriate changes, additions or explanations based on the reviewer's comments. It is recommended to receive.

7. PLOS authors have the option to publish the peer review history of their article (what does this mean?). If published, this will include your full peer review and any attached files.

Reviewer #1: No

Reviewer #2: No

---

## [Editor Report · Acceptance letter]

5 Apr 2024

PONE-D-23-37483R1 

PLOS ONE

Dear Dr. Miao, 

I'm pleased to inform you that your manuscript has been deemed suitable for publication in PLOS ONE. Congratulations! Your manuscript is now being handed over to our production team.

Kind regards, 

on behalf of

Dr. Philippe T. Georgel 

Academic Editor

PLOS ONE